



# Land-use perturbations in ley grassland decouple the degradation of ancient soil organic matter from the storage of newly derived carbon inputs.

Marco Panettieri[1,2], Denis Courtier-Murias[3], Cornelia Rumpel[4], Marie-France Dignac[4], Gonzalo Almendros[2], and Abad Chabbi[1,5]

[1]INRAE, AgroParisTech, UMR1402 ECOSYS, F-78850 Thiverval-Grignon, France
[2]Museo Nacional de Ciencias Naturales (MNCN-CSIC), c/Serrano 115-B, 28006, Madrid, Spain
[3]GERS-LEE, Univ Gustave Eiffel, IFSTTAR, F-44344 Bouguenais, France
[4]CNRS, IEES UMR (UPMC, CNRS, UPEC, IRD, INRAE)
[5]INRAE, UR P3F, 86600 Lusignan, France

*Correspondence to*: Abad Chabbi (abad.chabbi@inrae.fr)

**Abstract.** In a context of global change, soil has been identified as a potential carbon (C) sink, depending on land-use strategies. To detect the trends of carbon stocks after the implementation of new agricultural practices, early indicators, which can highlight changes in short timescales are required.

This study proposes the combined use of stable isotope probing and chemometrics applied to solid-state [13]C NMR spectra to unveil the dynamics of storage and mineralization of soil C pools. We focused light organic matter fractions isolated by density fractionation of soil water stable aggregates because they respond faster to changes in land-use than the total soil organic matter. Samples were collected from an agricultural field experiment with grassland, continuous cropping, and ley grassland under temperate climate conditions.

Our results indicated contrasting aggregate dynamics depending on land-use systems with grassland returning to soil larger amount of C as belowground inputs than cropping systems. Those fresh inputs are preferentially incorporated at the level of microaggregates, which are enriched in C in comparison with those of cropped soils. Land-use changes with the introduction of ley grassland provoked a decoupling of the storage/degradation processes after the grassland phase. The newly-derived maize inputs were barely degraded during the first three years of maize cropping, whereas grassland-derived material was depleted. As a whole, results suggest large microbial proliferation as showed by [13]C NMR under permanent grassland, then reduced within the first years after the land-use conversion, and finally restored. The study highlighted a fractal structure of the soil determining a scattered spatial distribution of the cycles of storage and degradation of soil organic matter related to detritusphere dynamics. In consequence, vegetal inputs from a new land-use are creating new detritusphere microenvironments rather than sustaining the previous dynamics, resulting in a legacy effect of the previous crop. Increasing the knowledge on the soil C dynamics at fine scale will be helpful to refine the prediction models and land-use policies.



## 1 Introduction

Soil carbon (C) stocks represent the largest C pool of the terrestrial biosphere (Scharlemann et al., 2014), which is accumulated or released to the atmosphere to an extent dependent on land-use and anthropogenic factors (Lal, 2004; Powlson et al., 2011). In fact, soil has the potential to store a large amount of C but also to emit great quantity of greenhouse gases (GHG) depending

on management practices (Lal, 2008; Smith, 2016). Agriculture is responsible for 20% of total GHG emission, but the transformation of soil into a C sink with sustainable agricultural practices (Chenu et al., 2019) has been proposed as a promising mitigation strategy by researchers, international panels and governments (IPCC, 2013; Lal, 2008; Minasny et al., 2017). These mitigation strategies need to be evaluated using adequate biomarkers that can decipher the stabilization/destabilization mechanisms and in particular the direction of change of the suitable land-use practices (Dignac et al., 2017; Wiesmeier et al.,

2019), and refine the prediction models about C balance associated with land-use policies (Chenu et al., 2019).

In this study we focused on the implementation of ley grassland rotations, which have been identified as a way to store carbon and provide ecosystem services (Kunrath et al., 2015; Lemaire et al., 2014). Nevertheless, few studies have been targeted on the long-term effects of ley grassland on soil organic matter (SOM) dynamics (Crème et al., 2018; Panettieri et al., 2017; Solomon et al., 2007). Changes in land-uses affect soil C stocks on a timescale of years or decades, therefore early

modifications in SOM dynamics may be undetectable when the quantification of soil C contents is performed on total soil rather than on reactive SOM pools (Castellano et al., 2015; Panettieri et al., 2017; Wiesmeier et al., 2019).

Assuming the above consideration, we hypothesized that the combination of two different land-uses (grassland and maize cropping) in a ley grassland rotation may produce contrasting effect on the dynamics of specific SOM pools. However, establishing an adequate experimental design to assess the effect of management practices on long-term C storage in the soil

is not a trivial task. Most of the research on soils conducted at the aggregate or molecular scales has been based on laboratory incubation conditions or with a limited experimental time (Dignac et al., 2017). Therefore, the extrapolation of those results to larger scales, longer intervals of time and more diverse soil conditions (land uses, physical and chemical characteristics) is always arbitrary. After nine years of ley grassland rotation the differences in C contents at the arable layer are only detectable at the aggregate scale, whereas bulk soil did not show significant change (Panettieri et al., 2017).

In consequence, the present research has been focused on the characterization of the light fraction of SOM (LF, particulate organic matter), i.e. a fraction that has been identified as an early indicator for changes in land-use (Courtier-Murias et al., 2013; Leifeld and Kögel-Knabner, 2005; Panettieri et al., 2014).

In a context of land-use change, a chemical characterization of SOM will establish C turnover rates and biochemical decomposition patterns, but the complex nature of SOM requires high-end analytical techniques (Derenne and Nguyen Tu,

2014), such as $^{13}$C nuclear magnetic resonance (NMR) or stable isotope probing (SIP). For soils that have experienced a land-use change with a conversion from C3 to C4 vegetation, the use of SIP is a valid method to measure the turnover of bulk SOM (Balesdent et al., 1987; Dignac et al., 2005) and specific SOM pools, such as LF-C (Bol et al., 2009; Matos et al., 2011;

Yamashita et al., 2006). The type of litter returned to soil is land-use specific and the litter quality may also affect the turnover rate measured for SOM pools (Armas-Herrera et al., 2016; von Haden et al., 2019). However, the type of litter and its

mineralization pattern are the two main proxies of SOM quality that can be assessed with solid-state 13C NMR (Baldock and Preston, 1995; Knicker et al., 2012). The use of 13C NMR on chemically or physically isolated SOM pools provides information on the previous land-use (Rabbi et al., 2014) and on the agricultural management (Panettieri et al., 2013, 2014).

The present study has been designed to identify the processes affecting the labile soil C pools resulting from changes in land-use. A temporary (ley) grassland system was compared to permanent grassland, permanent cropland and bare fallow soils as

controls using a novel approach based on combination of stable isotopes analysis and 13C NMR spectroscopy. To date, the combined use of SIP and 13C NMR on soil LF to assess the effect of changes of land-use on agricultural soils is scarce (Helfrich et al., 2006).

The hypothesis of this work is that the composition of SOM pools stored within different soil compartments may be used to obtain early information about the direction and magnitude of the change affecting SOM stocks in terms of accumulation or

mineralization, which depend on the litter nature (above vs. belowground biomass) and land-use characteristics (cropping vs. grassland). To test this hypothesis, the chemical composition of LF isolated from different water stable aggregates (Plaza et al., 2012) was characterized by solid-state 13C NMR spectroscopy. The obtained information was combined with measures of LF turnover in soil assessed by the natural abundance 13C enrichment of SOM provided by the in situ labelling of maize crops in a nine-year field experiment in western France.

**2 Materials and methods**

**2.1 Experimental area**

Soil samples were collected from the long-term experiment "Systems of Observation and Experimentation in Environmental Research-Agro-ecosystem, Biogeochemical Cycles and Biodiversity (SOERE-ACBB)" hosted at INRAE-Lusignan facilities (46°25′12.91″ N; 0°07′29.35″ E) in western France (Fig 1).

The pedoclimatic characteristics of the studied area have been extensively described elsewhere (Chabbi et al., 2009; Moni et al., 2010). In summary, the area has a temperate climate with around 846mm of annual precipitation, average annual temperature of 11.9 °C. The soil texture of upper soil horizons is a loamy, Cambisol (130 g kg-1 sand, 692 g kg-1 silt, 177 g kg-1 clay), while lower soil horizons are clayed rubefied with high content of kaolinite and oxides, classified as a Paleo-Ferralsol (103 g kg-1 sand, 612 g kg-1 silt, 286 g kg-1 clay). The soil bulk density was 1.48 g cm-3 (0-30 cm) with a pH(H2O) of 6.3 and 11

g kg-1 of organic carbon in the first 30 cm.





The long-term experiment started in 2005, on an area previously covered by oak forest and then devoted to agriculture or grassland for at least 100 years. The area was dominated by C3 vegetation and the soil δ13C signature at the beginning of the experiment was -25‰ relative to Vienna Pee Dee Belemnite (VPDB) standard.

A total of four treatments representing different land-uses were distributed on a 10 ha area with four replicates per treatment
arranged in four randomized blocks (one replicate per treatment in each block of about 4000 m2). Four different treatments were selected for sampling in the framework of the present experiment: (i) permanent crop rotations (PC), (ii) permanent grassland (PG), (iii) ley grassland (LG, 6 years of grassland followed by 3 years of continuous cropping), and (iv) bare fallow (BF). To take advantage of the *in situ* 13C labelling of SOM induced by maize plant inputs, only the subplots (500–700 m2) cultivated under maize (*Zea mays* L.) of PC and LG, were sampled (9 years under continuous maize for PC, and 6 years under
grassland followed by 3 years under maize, for LG). Grassland plots were sown with three dominant species *Lolium perenne* L. (cv Milca), *Festuca arundinacea* Scrheber (cv Soni), and *Dactylis glomerata* L. (cv Ludac) and hay was harvested and exported three times per year. Before each maize growing cycle, soil was tilled with a mouldboard plough at 25–30 cm depth yearly, followed by minor tillage operations before maize sowing (1 crop per year). All the treatments, except bare fallow subplots, were N fertilized. Grassland received between 170 and 380 kg N ha–1 year–1 (on average 240 kg ha-1 y-1), targeting
the nitrogen nutrition index (NNI) between 0.9 and 1 (Lemaire et al., 2008). Maize crops were fertilized following local agronomic practices and received between 36 and 160 kg N ha–1 year–1 (on average 98 kg N ha-1 y-1). Finally, subplots of bare fallow were 54 m2 wide without any input from vegetation or fertilization.

## 2.2 Sampling and aggregate fractionation

In July 2014, 20 cm diameter stainless steel cylinders were used to collect soil samples limiting aggregate disruption at a 0–30 cm depth, corresponding to the layer affected by tillage operations. Five soil cores were sampled for each plot at least 1 m far from the edges (three cores in the case of bare fallow subplots) and immediately merged to obtain a composite sample. Further information about the sampling procedure was reported in previous work (Panettieri et al., 2017).

Water content of bulk soil subsamples was determined gravimetrically right after the sampling, and then soil was dried at room
temperature before fractionation.

Water stable aggregates were isolated from bulk soil samples rewetted by slaking following the method of modified by Le Bissonnais (1996). Four soil aggregate fractions were obtained: larger macroaggregates (LMA, ∅ 2–7.1 mm), macroaggregates (MA, ∅ 0.2–2 mm), microaggregates (miA, ∅ 0.05–0.2) and silt + clay-size aggregates (S+C, ∅ < 0.050 mm). On average, the mass recovery was 98%, and not lower than 93% for all the samples. The mean weight diameter (MWD) for the four
treatments were calculated following the method of van Bavel (1950):

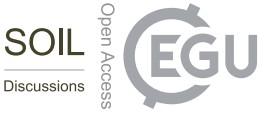

$$MWD = \sum_{i=1}^{n} \overline{D}_i \times f_i \tag{1}$$

In which *n* is the number of aggregate classes, $f_i$ is the relative abundance of the aggregate class, and $\overline{D}_i$ is the arithmetic mean between the upper and lower limit of the aggregate class.

**2.3 Isolation of light fractions of SOM**

The light fraction (LF) was isolated as the floating fraction during wet sieving for each aggregate sample, including the free and occluded sub-fractions. LF was extracted from soil samples using the method described by Kölbl and Kögel-Knabner (2004), modified to fit the experimental condition of this study. Briefly, 20 g of bulk soil samples or aggregate fractions were placed in a plastic vessel cooled by a water stream on external walls to dissipate the heat. Samples were dispersed in 200 mL of a sodium polytungstate (SPT, $3Na_2WO_4 \cdot 9WO_3 \cdot H_2O$, MW: 2986.01 g mol-1, Sigma-Aldrich) solution at a density of

1.8 g cm-3 using an ultrasonic probe (Scientific Bioblock Vibra-Cell 75115) calibrated to apply a power of 450 J mL-1, as described by Poeplau and Don (2014). After sonication, vessels were allowed to settle down overnight and total LF (free and occluded) was separated from the mineral phase by centrifugation at 1000 g. The LF samples were recovered using a pressure filtration system on a cellulose-free membrane filter (0.45 µm pore size Pall Life Science Supor® - 450) and successively washed with deionized water to remove all the SPT residues, until conductivity became lower than 5 µS cm-1. Finally, samples

were freeze dried and adequately stored.

For the present work, bulk soil samples from three blocks were analysed individually for each treatment (for a total of 12 bulk soil samples), whereas, for aggregate fractions, field replicates were merged in a composite sample (one aggregate fraction per treatment, for a total of 16 samples) to overcome constraints regarding quantity of fractions recovered and NMR instrumental time.

**2.4 Organic C and isotopic δ13C signature of samples**

The determination of total organic C (TOC), total N (TN) and 13C isotopic signatures of all the LF samples were performed on dry aliquots using an isotopic ratio mass spectrometer (VG SIRA 10) coupled to an elemental analyser (CHN NA 1500, Carlo Erba). The isotopic 13C/12C ratios (δ13C) were calibrated against the VPDB standard and expressed with equation (2):

$$\delta^{13}C = \left( \frac{(^{13}C/^{12}C)_{sample}}{(^{13}C/^{12}C)_{VPDB}} - 1 \right) \times 1000 \tag{2}$$

The turnover of LF carbon (LF-C) was quantified using the 13C enrichment induced by maize crops in plots under permanent cropland and ley grassland as described by Balesdent and Mariotti (1996), simplified as described by Dignac et al. (2005), equation (3):



$$F = \frac{LFC_{new}}{LFC} = \frac{\delta_{soilM} - \delta_{soilG}}{\delta_{newM} - \delta_{newG}} \qquad (3)$$

in which LFC refers to total C quantity within the soil LFC, and $LFC_{new}$ refers to C quantity within the LF-C derived from the
new maize vegetation.

For the isotopic ratios ($\delta$), the subscript *soilM* stands for the soil $\delta_{13}C$ measured for the two plots under maize (9 years of maize
for permanent cropland, and 3 years of maize after ley grassland), *soilG* stands for permanent grassland controls under
continuous C3 vegetation, newM and newG stand for the isotopic composition of maize and grass vegetal material. To estimate
the term ($\delta_{newM} - \delta_{newG}$), the difference in isotopic composition between plant materials corrected for above and below ground
inputs to soil was used, as calculated in a previous study on the same experimental farm (Panettieri et al., 2017). Using the
values of *F*, the percentage of C3-derived organic matter remaining in samples from ley grassland and permanent cropland
was calculated with reference to the permanent grassland samples. Similarly, degradation of LF in absence of new vegetal
inputs was calculated from bare soil samples. This approach produced an index of C3-LF persistence under different land-
uses.

**2.5 Solid-state $_{13}C$ Nuclear Magnetic Resonance**

The $_{13}C$ NMR analyses were carried out on a Bruker Avance 400 spectrometer operating at a $_{13}C$ frequency of 100.6 MHz
employing a $_{13}C$ ramped amplitude Cross Polarization Single Pulse sequence under Magic Angle Spinning conditions (Ramp-
CPSP/MAS). This sequence was first introduced by Shu et al. (2010) in material sciences, then successfully applied by
Courtier-Murias et al. (2014) on environmental samples. Spectra of LF samples obtained with a standard cross polarization
(Ramp-CP/MAS) sequence with an equal number of scans (i.e. the same acquisition time) were compared at the beginning of
the experiment (Supplementary Fig. S1). Preliminary experiment showed that Ramp-CPSP/MAS outperformed Ramp-
CP/MAS (CP, hereafter) in terms of signal-to-noise ratio by a factor of ca. ~2, in spectral regions with a lower proton density,
i.e. the aromatic region. Therefore, only Ramp-CPSP/MAS (CPSP, hereafter) analyses were carried out for the whole sample
set.

Approximately 50–100 mg of sample were placed into a zirconium oxide rotor with a diameter of 4 mm and sealed with Kel-
F® caps. For all the measurements, a spinning speed of 10 kHz was applied, the contact time was set to 1 ms and the recycle
delay was 3 s; this value was higher than in our previous works (Courtier-Murias et al., 2014) due to the technical specifications
of the NMR probe used in this study. About 5000–10000 scans were accumulated for each sample and a ramped $_1H$ pulse was
used during Hartmann-Hahn contact to circumvent Hartmann-Hahn mismatches. The spectra were divided in 8 main regions,
assignments for carbon resonances are reported in Table 1 according to Knicker and Lüdemann (1995) and Knicker (2011).



## 2.6 Statistical analyses

A Shapiro-Wilk test was used to check data normality before further analyses. For bulk soil and LF isolated from bulk soil samples, the significance of the differences found for the variables ($P \leq 0.05$) induced by the four land uses was assessed by non-parametric Kruskal-Wallis tests and the Dunn's multiple pairwise comparisons. For the NMR analyses, significance was assessed on bulk soil samples, whereas on composite samples, significance was not assessed due to the lack of replicate measurements. The significance level of Spearman's correlation coefficient ($\rho_s$) between the measured variables were assessed at a significance level of $P \leq 0.05$. A Principal Component Analysis (PCA) was used to explore how the chemical composition assessed by NMR affected sample distribution by groups. Since integrals of NMR regions are compositional data (their sum is the total measured intensity), data were pre-treated to reduce the effect of collinearity using the principles of Aitchinson's geometry and center logratio transformed (CLR) prior to the PCA calculation (Aitchison, 1982). Total organic carbon (TOC), carbon to nitrogen ratios (C/N) and C3-derived LF-C losses were used as supplementary variables. Statistical analyses were carried out using XLSTAT (Addinsoft, Boston, USA, https://www.xlstat.com).

A chemometrics approach was used to treat NMR spectra and obtain information about the relative contribution of C3 and C4-derived organic matter to LF fractions. Spectra were exported as a 2-column matrix reporting chemical shift (516 points corresponding to about 2 points per ppm) and absolute intensity for each point. Afterwards, normalized intensities ($I_f$) were calculated to overcome the different C contents of each sample that may lead to different signal to noise ratios for each spectrum following equation (4) and (5).

$$I_t = \sum_{n=1}^{516} I_n \qquad (4)$$

$$I_f = I_n \div I_t \times 1000 \qquad (5)$$

Measured intensities at each point ($I_n$) were divided by the total spectrum intensity ($I_t$) calculated as the sum of the intensities for each point ($n= 516$) then multiplied by an arbitrary factor of 10000 to keep the normalized intensities within a range of -2 to 40.

Normalized intensities were used to obtain scaled spectra, one for each sample. Afterwards, the samples of permanent grassland were chosen as reference spectra (free of C4-derived SOM and with continuous vegetal inputs) whereas spectra from other treatments were subtracted from the corresponding permanent grassland sample spectra. This leads to a graphical view on changes in LF originated by different land uses, with positive signal for the regions in which permanent grassland had higher relative contribution than the subtracted treatment, and negative signal for the opposite situation.



## 3 Results and discussion

### 3.1 Carbon contents in total soil and light fractions

The TOC contents and the relative contribution of the LF for the four treatments are represented as histograms in Fig. 2.
The LF-C contents represented between 7 and 30% of the TOC for bulk soil and aggregates, in line with results of other studies
(Leifeld and Kögel-Knabner, 2005). Samples from permanent cropland showed the higher contribution of LF to total stocks
of C among the four treatments, whereas larger macroaggregates showed the highest LF-C contribution among the fractions
(Fig. 2). The LF-C contents decreased under ley grassland if compared with permanent grassland, and permanent cropland for

the bulk soil, the larger macroaggregates and the microaggregates fractions (Fig. 2). Bearing in mind the relative contribution
of LF-C to TOC, LF-C has been proposed as an early indicator of changes affecting soil quality due to the faster turnover than
TOC, normally on a timespan of years (Poeplau et al., 2018). This useful characteristic has been proposed to detect changes
in C stocks modulated by land-use against the large background of TOC that is not affected (Leifeld and Kögel-Knabner,
2005). Our results confirm the fast response of LF-C to land-use. While no significant differences between vegetated treatments

were found for TOC contents (Panettieri et al., 2017, Fig 2), carbon of aggregate fractions showed significant changes. The
LF-C contents under ley grassland were similar to those measured for bare soil. We attribute these results to the effect of soil
perturbation due to the switch from a grassland soil ecosystem to a crop soil ecosystem. Deep tillage operations, as performed
during maize cropping, are associated with LF-C degradation due to the increase of microbial activity (Courtier-Murias et al.,
2013; Panettieri et al., 2014). Soil aggregation responds quickly to changes in land-use (Álvaro-Fuentes et al., 2008; Bronick

and Lal, 2005) and stable aggregates provide a protective environment for land-use specific proportions of LF-C in soil (Leifeld
and Kögel-Knabner, 2005). Another important aspect to consider in soils subjected to land-use change is the different types of
vegetal inputs: grassland is characterized by an extended, dense and relatively shallow root system, whereas maize roots are
more spaced and deep (Jackson et al., 1996). In this experiment, grassland total inputs to soil were higher than those from
maize crops, but the maize returned more inputs from shoots whereas the grassland inputs to large extent consisted in root-

derived material (Panettieri et al., 2017). On one hand, maize provides a large proportion of inputs of aboveground biomass
successively incorporated into soil during tillage, and a lower percentage of belowground inputs as root-derived material. On
the other hand, grassland provides a large proportion of belowground inputs to soil in a more extended area. The C-to-N ratios,
$\delta_{13}C$ signature and MWD of the samples are reported in Table 2. Despite the higher number of tillage operations performed
under ley grassland and permanent cropland, the MWD was higher for those two treatments if compared with permanent

grassland and bare fallow soils. Thus, under the conditions of the studied area, maize cropping provides larger aggregation
strength than permanent grassland. Given that grassland returns to soil a larger amount of belowground inputs than maize
crops, we can suggest that such incorporation takes place at the level of smaller size aggregates richer in C, in comparison with
the incorporation of C originating from maize crops.



### 3.2 Local proxies of soil organic matter dynamics

The percentages of newly C4-derived LF-C for permanent cropland and ley grassland are shown in Fig. 3. Permanent cropland showed higher proportions of C4-derived LF-C than ley grassland within all the aggregate fractions, due to the longer time cropped under maize. LF-C from silt and clay fraction was mostly composed of C3-derived C, less than 5% of new C was found for permanent cropland and no new inputs were detected for ley grassland. The contribution of new C in the LF increased with aggregate size for all the fractions of permanent cropland; 31% of LF-C in LMA of permanent cropland was maize-

derived, evidencing the faster turnover of larger aggregates, which has been extensively described in the literature and correspond to the preferential accumulation of particulate, slightly decomposed LF in coarse soil fractions (Puget et al., 1995, 2000; Tisdall and Oades, 1982). A different trend was observed for ley grassland treatment, the contribution of new C to the LF of LMA was similar to that of permanent cropland. However, the contributions of maize-derived C to the LF-C of MA and MiA of ley grassland were very similar, meaning that the increase of C4-derived LF-C with the size of aggregates did not

follow the linear pattern for MA found in ley grassland (Fig. 3).

Taking into account the total amount of LF-C (C3 and C4-derived), Fig. 4 shows how permanent cropland LMA and MiA incorporated a greater amount of LF-C than the same fractions of permanent grassland. The LMA and MiA fractions of ley grassland contained the lowest amount of LF-C if compared with all the treatments.

The results also showed how a three-year continuous maize cropping following a six-year grassland produced severe disruption

and/or rearrangement of C pools. The measured losses of LF-C under ley grassland were not supported by similar losses of TOC, therefore the tillage operations and maize cropping may have led to redistribution of this C, favouring its incorporation into heavier mineral associated C-pools (Basile-Doelsch et al., 2009). In fact, the observed trends for soils under permanent maize suggested that longer time of maize cropping will restore LF-C of LMA and MiA. However, losses of C3-derived LF-C were not registered for plots under permanent maize. Only two pools of LF-C have been partitioned using the $_{13}$C *in situ*

labelling, but we cannot exclude that the C3-derived LF-C is on its turn composed of different land-use specific pools accumulated before the establishment of the experiment (DeGryze et al., 2004; Meyer et al., 2012). Those pools may have been more susceptible to alteration by the alternation of grassland and maize, but not to continuous maize cropping.

Since no further isotopic partitioning is possible on the LF-C accumulated before the beginning of the experiment, chemical composition of LF-C pools will be used to provide further insights about the effect of land-use change on C stocks.

### 260 3.3 Performance of the NMR method

To test the performance of CPSP sequence against the more commonly used CP, two spectra from the same sample were acquired using the same number of scans for both experiments. Figure S1 shows that CPSP sequences was able to detect a higher intensity of the signal for aromatic-C in comparison with CP sequence, with negligible modifications detected for the other regions. This is due to the lower proton density in condensed aromatic moieties, that can lead to a less effective



polarization transfer from proton to carbon nuclei. In CP experiments, an increase of the contact time to transfer the magnetization along the distance between condensed aromatic-C and closest proton could be used as a solution to overcome this problem. However, longer contact times are also correlated with losses of signal intensity mediated by the spin-lattice relaxation, which will produce lower signal intensity (Knicker, 2011). The extra $^{13}$C pulse of the CPSP sequence allows to better measure $^{13}$C atoms of the condensed aromatic moieties that are far from protons and therefore improving the signal-to-

noise ratio of this region especially when their $^{13}$C NMR $T_1$ relaxation values are short (Courtier-Murias et al., 2014). In addition, some differences for $CH_2$ groups in non-crystalline poly(methylene) and carbonyls groups have also been detected (Courtier-Murias et al., 2014). However, CPSP always equals or improves CP performance even for soils with a low aromaticity, as confirmed for our comparison spectra. In consequence, CPSP was therefore selected as the standard sequence for this study.

Standard deviations of the calculated areas for the field replicates (three different blocks) of bulk soil samples were lower than 1.35% for all the integrated regions, except for *O*-alkyl C region of permanent grassland (2.05 %). This assessed that the variability due to spatial conditions and sample preparation was reasonably low and the integrated areas of the spectra from the composite samples are valid to be interpreted in terms of differences in SOM in the different aggregate fractions.

Contributions in carboxyl C and *N*-alkyl C were constant (7–9 % and 9–10 %, respectively) for all the measured samples, with

the exception of bare fallow samples, in which carboxyl C accounted for more than 10% in LMA and MA. The region assigned to *N*-alkyl C may also represent the typical signal assigned to methoxyl C of lignin structures (Lüdemann and Nimz, 1973). Signal intensity in the *N*-alkyl region showed a significant positive Spearman's correlation with total N content ($\rho_s$= 0.647, $P$ < 0.05) and a significant negative correlation with the intensity of the heteroaromatic C region ($\rho_s$= -0.574, $P$ < 0.05), suggesting that the *N*-alkyl signal is derived mainly from C in proteinaceous material rich in N rather than methoxyl C. The

persistence of protein derived material in SOM pools has been described in other studies (Diekow et al., 2005; Nannipieri and Eldor, 2009; Panettieri et al., 2014) and it has been used to characterize LF as "new" SOM, rich in fresh litter, but also exoenzymes and cytoplasmic material from microbial biomass and necromass (Miltner et al., 2012).

The alkyl-to-*O*-alkyl C ratio is commonly used as a proxy of SOM degradation (Baldock and Preston, 1995). Isotopic results showed how larger aggregates contain fresher LF-C, confirmed by the decrease of the alkyl/*O*-alkyl with the increase of

aggregate size. Fresh litter is richer in carbohydrates from cellulose which anomeric C resonates in the *O*-alkyl region, whereas litter in a comparatively more advanced stage of degradation is characterized by the progressive enhancement of alkyl C signal suggesting the selective enrichment of long-chain and condensed aliphatic structures, including cutins and suberins from higher plants, or phospholipids from microbial and fungal biomass (Miltner et al., 2012; Panettieri et al., 2013). In this study, the presence of a clear peak of terminal methyl group that accounted for half of the intensity of the methylene group indicated that

most of the total alkyl C intensity is due to chains shorter than those expected for cutins and suberins. Alkyl C contribution was higher for smaller fractions, probably explained for this experiment by the microbial growth stimulated by the diffusion





into fine pores of smaller molecules released during the enzymatic breakdown of macromolecules and by the adsorption of patchy fragment of necromass membrane and plant aliphatic material on mineral and organic surfaces (Basile-Doelsch et al., 2015; Leifeld and Kögel-Knabner, 2005; Ludwig et al., 2015).

The LMA fractions of the three treatments with vegetal inputs were characterized by a high contribution of *O*-alkyl C; in the case of temporary grassland, 43% of the total C intensity measured by NMR was assigned to carbohydrates, a value close to that for the non-degraded plant tissue. Fig. 5 shows the alkyl/*O*-alkyl ratios for the four treatments. Trends along the aggregate fractions evidenced how ley grassland had similar ratios than those of permanent grassland for fine fractions, but a totally different ratio for LMA.

At this point, the interpretation of the relative abundances of each compound classes combined with the subtraction spectra constitutes a valid and original approach to understand how land-use affects the dynamics of SOM pools within different aggregate compartments, using the *in-situ* labelling provided by maize as a benchmark to quantify the proportion of LF-C that has changed during the experiment.

**3.4 Chemical composition of the soil organic matter pools from different land-uses**

A PCA analysis was performed to represent the differences of LF isolated from aggregate fractions of four treatments based on the relative abundance of NMR compound classes (Fig. 6). In the plane defined by the two first components (82.5% variance explained), the four treatments could be differentiated along the first component, while the second component ordered the samples by aggregate size. The scores corresponding to permanent grassland fractions were clearly placed on the left of the plot, on the opposite side of the bare fallow ones. Permanent cropland fractions were partially overlapped to the area described

by bare fallow, and by temporary grassland. The LMA fractions were positioned on the top of the plot, finer fractions on the bottom. The ley grassland treatment presented the greatest scattering along the second component and the thinnest scattering along the first one. Two out of four fraction classes (MA and S+C) may be connected with a quasi-horizontal line from left to right following the order PG–TG–PC–BS, whereas LMA and MiA had a more scattered distribution. Looking at the position of the different treatments, chemical composition of LF-C from ley grassland was similar to that of permanent cropland, and

evidencing a prominent shift from the original grassland footprint to the cropland footprint. Similar changes occurred in the molecular composition of SOM studied by analytical pyrolysis (Rumpel et al., 2009). This result evidenced that $^{13}$C NMR analyses of LF-C may be useful to detect changes in SOM quality due to land-use on a short-term timescale, since this shift of ley grassland samples through a permanent cropland footprint was not detected with analyses performed on total SOM on the same experimental area (Crème et al., 2018; Panettieri et al., 2017).

The active variables of the PCA were split in two groups; on the right part, a cluster formed by the variables aromatic, *O*-aryl and carboxyl C was correlated to the first component. The observations under bare fallow had higher relative contributions from these three regions. This cluster can be interpreted as an advanced status of degradation of the organic matter in the LF



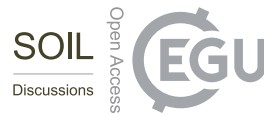

associated with the land-use (Leifeld and Kögel-Knabner, 2005). On the left and more scattered along the second component axis, the variables *O*-alkyl, *N*-alkyl, alkyl and terminal alkyl C, were more correlated with the degradation status of LF-C
within the different aggregate sizes.

The exploration of PCA indicated that the type of land-use lead to the highest distances for homologous LMA and MiA fractions, showing that the chemical differences originating from by the four land uses within those fractions had the larger magnitudes. The supplementary variables were most effectively described by the 13C NMR regions *O*-alkyl, *N*-alkyl and terminal alkyl, plus the cluster of degradation status on the right side of the plot.

The supplementary variables TOC and C/N were placed on the left part of the graph, TOC was correlated to the area described by the permanent grassland observation and the *N*-alkyl C, whereas C/N was correlated to the LMA area and *O*-alkyl C. Losses of C3 derived LF-C were placed on the right of the plot, closely correlated to the area of the graph described by bare soil and perfectly opposite (thus negatively correlated) to *N*-alkyl C and terminal alkyl. The fact that mineralization of LF-C from previous land-use was correlated to the N cycle (in this case, the 13C NMR signal attributed to proteinaceous material) and to
the intensity of the terminal methyl group attributed to microbial aliphatic material agrees with recent findings about the stoichiometric relationships controlling the microbial degradation of vegetal litter (Chen et al., 2019; Sinsabaugh et al., 2013).

**3.5 The effect of land use on the degradation status of organic matter pools**

The spectral subtractions with respect to permanent grassland samples were used to define selective losses (positive values) or gains (negative values) in the 13C NMR intensities for each land-use (Fig. 7).

The results highlighted a higher contribution of *O*-alkyl C in the LF isolated from bulk soil and most of the aggregate fractions of permanent grassland than in the other treatmens, with the exception of LMA in temporary grassland and permanent cropland. This trend was compensated with an enrichment in the aromatic, heteroaromatic and carboxylic regions with respect to the intensities registered for permanent grassland soils. With the exception of LMA fractions, the magnitude of these effects increased from ley grassland to permanent cropland to bare fallow, clearly indicating that LF-C of the treatments under maize
presented a more degraded status than those under permanent grassland. The LF-C isolated from silt and clay fraction of permanent grassland had also higher intensities in alkyl and terminal alkyl regions. Those trends could be explained according to the SOM degradation pattern registered for bare fallow soils. When a soil under grassland is left bare for nine years, the quantity of LF-C decreased (presumably lost following microbial degradation, rather than from translocation to mineral-associated fraction) and 13C NMR intensity assigned to carbohydrates of LF-C decreased with the size of the aggregates
(Panettieri et al., 2014; Plaza et al., 2013; Six et al., 2004). The data also suggest that the LF-C of the silt and clay fraction suffers minor changes related to land-use in terms of quantity, but the 13C NMR signal intensity attributed to alkyl decreases for this fraction when soil is either cultivated under maize or left bare. This change is mostly attributable to the losses of microbial-derived C presumably rich in short aliphatic chains, as explained above.



The presence of higher proportion of carbohydrate-derived material in LF-C isolated from large aggregates and the
corresponding higher contribution of microbial C in fine fraction agrees with the literature describing the size-dependant
reactivity of the aggregates (Puget et al., 1995; Six et al., 2000; Tisdall and Oades, 1982). Large aggregates contain high
amounts of fresh plant material rich in carbohydrates that is preferentially degraded by exoenzymes (Baldock and Preston,
1995), the so-called detritusphere. Moreover, aggregates contain the by-products of the enzymatic breakdown of
macromolecules, which tends to diffuse into finer pores, sustaining the higher microbial proliferation in finer fractions
(Courtier-Murias et al., 2013; Ludwig et al., 2015; Miltner et al., 2012). When not enough macromolecules are degraded in
coarser fractions (i.e. lower carbohydrate contribution), this proliferation is not sustained anymore (Plaza et al., 2013).
However, when a different vegetal input is returned to the soil after changing land-use (from grassland to permanent maize
cropping), losses in alkyl C intensity of LF-C for silt and clay fraction were also registered, indicating that the location and the
type of inputs had an influence on the alkyl C contribution to this fraction (Eclesia et al., 2016). After three years of maize
cropping, LF-C from ley grassland showed a higher proportion of carbohydrate derived material (*O*-alkyl) in the LMA and a
higher contribution of maize-derived LF-C, as assessed by isotopic analyses. Therefore, land-use specific characteristics of the
maize phase such as the new type of vegetal input, a different root network or the tillage operation were responsible of a change
in quantity and chemical composition of the LF-C from LMA under ley grassland (von Haden et al., 2019)

For LF-C extracted from the other fractions of ley grassland, the *O*-alkyl intensities were lower than those of permanent
grassland. A similar trend of *O*-alkyl losses in comparison to permanent grassland was detected for permanent cropland, with
the exception of the LMA fraction in which the *O*-alkyl intensities were similar in the two treatments. Therefore, we can infer
that (i) maize inputs rich in carbohydrates are mostly deposited as surface litter (Panettieri et al., 2017), (ii) maize cropping
increases soil MWD compared to grassland, and (iii) surface litter lasts in LF-C of LMA as non-degraded material three years
after land-use change, but a degradation and/or a redistribution to heavier soil fraction is expected on a time scale of nine years.
On the other hand, grassland provides higher belowground inputs resulting in higher amounts of LF-C but in this case lower
MWD. These results may be explained by belowground inputs rich in vegetal macromolecules, which are progressively
degraded and redistributed from larger to finer aggregates. They may be feeding and sustaining the microbial growth found in
the finest fraction of permanent grassland. As soon as grassland is substituted by maize, this flux of nutrients is hampered and
the contribution of microbial C to LF-C decreases. Another type of aboveground input is returned to the soil and left
"untouched" until a new type of detritusphere ecosystem is built around it and the new equilibrium will be reached (Kumar et
al., 2016; Kuzyakov and Blagodatskaya, 2015). In fact, the *O*-alkyl C contribution to silt and clay fraction of permanent
cropland is higher than that found for ley grassland, demonstrating that the 13C NMR signal attributed to carbohydrates is
restored within a longer timescale. The coarser nature of maize-derived aboveground inputs in comparison with grassland
belowground ones may also explain the trend to higher MWD when maize is implemented into the crop rotation.





Comparison of the trends of the three vegetated treatments with those found for bare fallow shows that, despite the new inputs returned to soil under maize, the change in land-use in ley grassland will provoke the disruption of the equilibrium reached under the previous grassland cover. Some of the aggregate fractions of ley grassland will continue to function if the land-use change did not happen, showing a degradation of LF-C in terms of quantity of quality similar to that registered for bare fallow. This may be explained by the fact that land-use specific microenvironments are not adapted to the degradation of the new

vegetal inputs because of its different chemical characteristics and/or different spatial arrangement (Castellano et al., 2015; Eclesia et al., 2016). Later on, the newly built aggregates and microenvironments under the new phase of the rotation will become the majority of the total soil matrix, switching the net soil functionality to the new land-use. We suggest that this may be one of the explanations for the so-called "legacy effect", described as the influence of previous land-uses on the soil C recovery/loss dynamics after the establishment of a new soil management (Compton et al., 1998; Smith, 2014). Moreover, the

optimum for soil microbial diversity and soil C storage tends to happen at an optimum level of soil perturbation and land-use switch following a humped back curve, different for each type of soil (Acosta-Martínez et al., 2008; Tardy et al., 2015). For our study, three years of maize cropping in ley grassland affected the LF-C dynamics in a way similar to that observed for bare fallow. This is clearly an alert signal that C stocks accumulated under grassland may be hampered by future cropping years (Sleutel et al., 2006). Nevertheless, C losses were not observed for bulk soil (Crème et al., 2018) and longer times under maize

tended to restore LF-C pools and increase MWD, but with an overall loss of C stocks (Panettieri et al., 2017). Therefore, refining the data on the land-use depending C persistence in soil may be helpful to decide which land-use rotations will be the most suitable for C storage strategies (Rumpel et al., 2019).

## 4 Conclusions

This study proposes a novel approach to unveil the land-use dependency of the storage and degradation dynamics that affect

reactive C pools. Our findings indicate that C under ley grassland is subjected to two different and mostly independent mechanisms: the degradation regarding the grassland-derived LF-C and the accumulation of new maize-derived LF-C. Considering the difference between grassland species and maize plant, we assume that root architecture of the rhizosphere contributed to the change in the chemical nature and spatial distribution of the vegetal inputs returned to soil.

We found evidences that these factors regulated by land-use led to the formation of land-use specific detrituspheres in ley

grassland, each of them with specific LF-C dynamics of redistribution among different pools and mineralization. The microbial proliferation suggested by $^{13}$C NMR for LF-C accumulated during the grassland phase is not sustained during the maize phase, as if the microenvironments and microbial communities were not sensitive to the new maize inputs returned as coarse material. As a whole, results showed that grassland derived LF-C continues to be degraded as if no inputs were returned to soil, whereas maize-derived material is slowly degraded. We expect that longer maize cropping time will establish a new equilibrium among



LF-C isolated from aggregates. Agricultural intensification in ley grassland provokes firstly a decrease of soil LF-C, then a depletion of total soil C contents. The analytical characterization of LF-C is here proposed as a way to evaluate the impact of crop rotations at shorter timescale, before that soil C contents are hampered. This study enables to generate sufficient evidence and understanding of C dynamics at fine scale to devise SOC model predictions and policies to sustain C storage under land use practices.

**Figure captions**

**Figure 1.** Lusignan national long-term observatory at the Nouvelle-Aquitaine region France (a); Land-use management of target treatments from 2005-2013 used in this study before the sampling (b). Continuous maize bands were installed in subplots of the T1 and T3 treatments, in addition to bare soil that have not received fresh organic matter input since the start of the experimentation in 2005.

**Figure 2.** Light fraction carbon (LF-C) contribution to total organic carbon in bulk soil (n=3) and density fractions (n=1) for different treatments (PG=permanent grassland, PC = permanent cropping, TG= temporary grassland, BS = bare soil). No significant differences between treatments ($P < 0.05$) for bulk samples were found. Error bars show the calculated standard deviations for replicate samples.

**Figure 3.** Percentage of maize-derived (C4) light fraction carbon (LF-C) for bulk soils and aggregate fractions under permanent 435 cropland and temporary grassland. LMA: larger macroaggregates; MA: macroaggregates; MiA: microaggregates; S+C: silt and clay. * values of S+C for temporary grassland were slightly lower (-0.4%) than those found for permanent grassland, probably due to field variability. Error bars show the calculated standard deviations for replicate samples.

**Figure 4.** C3 and C4 light fraction C contribution to total light fraction C for different treatments and soil fractions. PG: permanent grassland; TG: ley (temporary) grassland; PC: permanent cropland; BS: bare fallow soil; LMA: larger 440 macroaggregates; MA: macroaggregates; MiA: microaggregates; S+C: silt and clay. No significant differences between treatments ($p < 0.05$) were found for bulk soil samples. Error bars show the calculated standard deviations for replicate samples.

**Figure 5.** The alkyl to *O*-alkyl ratios calculated for light fraction-C isolated from bulk soils and aggregate fractions for the four treatments. LMA: larger macroaggregates; MA: macroaggregates; MiA: microaggregates; S+C: silt and clay.

**Figure 6.** Results of principal component analysis applied to 13C NMR analysis the of C distribution within different aggregate 445 fractions from soils under different land use. Projected loadings of the soil measured variables (left) and representation of the

light fraction C isolated from aggregate-size fractions of the four different treatments on the plain defined by the two first principal components (right). Labels for samples and variables refer to Tables 1, 2 and 3.

**Figure 7.** Comparisons of $^{13}$C NMR spectra based on subtractions of the spectra obtained from temporary grassland (TG), permanent cropland (PC) and bare fallow (BS) samples from the homologous spectra obtained from permanent grassland (PG, first row, used as controls). Spectral regions: carboxyl C (R1), heteroaromatic (R2), aromatic (R3), anomeric (R4), *O*-alkyl (R5), *N*-alkyl (R6), alkyl (R7), terminal alkyl (R8). For graphical reasons, all the intensities of all the resulting spectra were normalized by an arbitrary factor (10000) to fit within the interval of -2 to +40 arbitrary units. The y axis of all the graphs was scaled to the same interval of arbitrary units, except for LMA of temporary grassland.

**Figure S1:** Comparison of $^{13}$C NMR spectra from Ramp-CPSP/MAS and Ramp-CP/MAS sequences on soil light fractions. The region assigned to aromatic-C had higher intensity with the Ramp-CPSP/MAS sequence.

**Authors' contribution**

Based on the CASRAI's CRediT definitions of contributor roles, the authors contributed to this work as follow. *Conceptualization:* MP, AC, CR, MFD. *Data curation:* AC, CR, MP, GA. *Formal analysis***:** MP, DCM, MFD, CR, GA. *Funding acquisition:* AC, CR, MP. *Investigation:* MP, DCM. *Methodology:* MP, DCM, CR, MFD. *Project administration:* AC, MP, CR. *Resources:* AC, CR, MFD, GA. *Software:* DCM, GA, MP. *Supervision:* MP, CR, AC. *Validation:* AC, CR, MFD. *Visualization:* MP, GA. *Writing – original draft:* MP. *Writing – review & editing***:** DCM, GA, MFD, CR, AC.

**Data availability**

The data generated in the framework of the SOERE-ACBB observatory are freely available and accessible. The data that support the findings of this study are available from the corresponding and lead authors, upon reasonable request.

**Acknowledgments**

The authors acknowledge the support of the AgreenSkills fellowship programme which has received funding from the EU's Seventh Framework Programme under grant agreement n° FP7-26719, the AnaEE France - grant agreement n° ANR-11-INBS-0001 and the CNRS-INSU. The lead author Marco Panettieri acknowledges the funding received through the Transnational Access to Research Infrastructures activity in the 7th Framework Programme of the EC under the ExpeER project n° 262060 for conducting the research.



Financial support from the IR-RMN-THC Fr3050 CNRS for conducting the research is gratefully acknowledged. We are deeply indepted to Xavier Charrier for his technical support at the experimental area SOERE-ACBB (Systems of Observation and Experimentation in Environmental Research- Agro-ecosystem, Biogeochemical Cycles and Biodiversity), Valerie Pouteau, Cyril Girardin, and Daniel Billiou for their technical support in the analyses carried out.

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

**Tables**

**Table 1.** Typical assignments for peaks in 13C NMR solid-state spectra from geochemical samples (Knicker, 2011; Knicker and Lüdemann, 1995). Reference used:  Tetramethylsilane = 0 ppm.

| Chemical shift range (ppm) | Name of the spectral region | Assignment |
|---|---|---|
| 210-160 | *Carboxyl* | Carboxyl, carbonyl, ester, and amide carbons. |
| 160 – 140 | *Heteroaromatic* | Aromatic COR or CNR groups, furans. |
| 140 – 110 | *Aromatic* | Aromatic C-H carbons, guaiacyl C-2 and C-6 in lignin, olefinic carbons, bridging C in polyaromatic units. |
| 110 – 90 | *Anomeric* | Anomeric carbon of carbohydrates, syringyl C-2 and C-6 in lignin. |
| 90 – 60 | *O-alkyl* | Carbohydrate-derived structures (C-2 to C-6) in hexoses, C-α of some amino acids, higher alcohols. |
| 60-45 | *N-alkyl* | Methoxyl groups, C-α of most amino acids, *N*-alkyl C. |
| 45 – 25 | *Alkyl* | Methylene groups in aliphatic rings and chains. |
| 25-0 | *Terminal alkyl* | Terminal methyl groups. |






**Table 2.** C-to-N ratios, mean weight diameter (MWD) and δ13C signature measured for the bulk soil and aggregate fractions of four treatments. Significant differences between treatments ($P < 0.05$) for bulk soil samples are indicated with different letters.

| Treatment | Fraction | C/N | Mean weight diameter (MWD) mm | δ13C signature ‰ |
|---|---|---|---|---|
| *Permanent cropland* | Bulk soil | 14.0 ± 2.0 | | -25.0 ± 0.02 **b** |
| | LMA | 18.4 | | -22.9 |
| | MA | 15.1 | 0.70 ± 0.17 **b** | -24.9 |
| | MiA | 13.5 | | -25.4 |
| | S+C | 15.3 | | -26.3 |
| *Ley grassland* | Bulk soil | 13.9 ± 0.9 | | -26.6 ± 0.2 **ab** |
| | LMA | 20.0 | | -23.6 |
| | MA | 18.2 | 0.66 ± 0.16 a**b** | -26.1 |
| | MiA | 13.1 | | -25.9 |
| | S+C | 14.7 | | -27.0 |
| *Permanent grassland* | Bulk soil | 13.7 ± 0.7 | | -27.5 ± 0.2 **a** |
| | LMA | 14.8 | | -27.7 |
| | MA | 16.4 | 0.54 ± 0.16 **ab** | -27.6 |
| | MiA | 14.3 | | -27.4 |
| | S+C | 13.7 | | -26.9 |
| *Bare fallow* | Bulk soil+ | 14.7 ± 1.6 | | -26.9 ± 0.2 **ab** |
| | LMA | 10.6 | | -25.5 |
| | MA | 12.5 | 0.40 ± 0.10 **a** | -26.4 |
| | MiA | 15.3 | | -27.0 |
| | S+C | 11.8 | | -26.6 |




**Table 3.** Integration values (expressed as a percent of the total spectral area), and signal-area ratios, of the main regions of $^{13}C$ NMR spectra from soils under contrasted agricultural management ± standard deviations. Significant differences between treatments ($P < 0.05$) for field replicates of bulk soil samples ($n=3$) are indicated with different letters.

| | | Carboxyl | Hetero-aromatic | Aromatic | Carbohydrates Anomeric | Carbohydrates O-Alkyl | N-Alkyl or Methoxyl | Alkyl Methylene | Alkyl Methyl | Carbohydrates TOT | Alkyl TOT | Aryl/O-Aryl |
|---|---|---|---|---|---|---|---|---|---|---|---|---|
| Permanent Cropland | Bulk | 8.96 ± 0.94 | 7.76 ± 1.02 | **21.91 ± 1.03 ab** | 8.77 ± 0.13 | 24.07 ± 1.35 | 9.29 ± 0.55 | 13.10 ± 0.81 | 6.14 ± 0.61 | 32.84 ± 1.22 | 19.24 ± 1.30 | 2.85 ± 0.24 |
| | LMA | 6.97 | 7.32 | 19.99 | 9.84 | 27.45 | 9.24 | 12.68 | 6.50 | 37.29 | 19.18 | 2.73 |
| | MA | 8.28 | 7.34 | 21.86 | 8.98 | 25.16 | 9.54 | 13.02 | 5.81 | 34.14 | 18.83 | 2.98 |
| | MiA | 8.00 | 7.37 | 21.65 | 8.61 | 24.46 | 10.10 | 13.31 | 6.49 | 33.07 | 19.80 | 2.94 |
| | S+C | 8.75 | 7.24 | 21.94 | 8.03 | 22.52 | 9.69 | 14.90 | 6.93 | 30.55 | 21.83 | 3.03 |
| Ley Grassland | Bulk | 8.25 ± 0.39 | 7.26 ± 0.20 | **21.56 ± 0.62 ab** | 8.54 ± 0.37 | 24.43 ± 0.54 | 9.76 ± 0.39 | 13.61 ± 0.13 | 6.60 ± 0.67 | 32.97 ± 0.21 | 20.21 ± 0.78 | 2.97 ± 0.08 |
| | LMA | 6.96 | 5.93 | 17.78 | 9.66 | 33.72 | 9.28 | 10.65 | 6.02 | 43.38 | 16.67 | 3.00 |
| | MA | 7.95 | 7.09 | 21.60 | 8.67 | 26.60 | 9.57 | 12.61 | 5.90 | 35.27 | 18.51 | 3.05 |
| | MiA | 8.51 | 7.63 | 21.52 | 9.09 | 24.96 | 9.46 | 12.51 | 6.32 | 34.05 | 18.83 | 2.82 |
| | S+C | 7.92 | 7.28 | 21.56 | 7.84 | 21.62 | 9.40 | 16.35 | 8.02 | 29.46 | 24.37 | 2.96 |
| Permanent Grassland | Bulk | 8.51 ± 0.37 | 7.16 ± 0.39 | **20.86 ± 1.08 a** | 8.45 ± 0.24 | 25.63 ± 2.05 | 9.74 ± 0.25 | 13.12 ± 0.52 | 6.53 ± 0.86 | 34.08 ± 1.87 | 19.64 ± 1.21 | 2.91 ± 0.06 |
| | LMA | 6.90 | 6.32 | 19.77 | 9.34 | 28.48 | 9.79 | 12.81 | 6.59 | 37.82 | 19.40 | 3.13 |
| | MA | 7.27 | 5.72 | 18.51 | 8.63 | 29.65 | 9.83 | 12.76 | 7.63 | 38.28 | 20.39 | 3.24 |
| | MiA | 7.99 | 6.75 | 19.47 | 8.37 | 26.57 | 10.49 | 13.11 | 7.24 | 34.94 | 20.35 | 2.88 |
| | S+C | 7.76 | 6.70 | 20.18 | 7.77 | 22.48 | 9.78 | 16.21 | 9.13 | 30.25 | 25.34 | 3.01 |
| Bare Fallow | Bulk | 7.93 ± 0.75 | 7.75 ± 0.31 | **22.71 ± 0.15 b** | 8.71 ± 0.22 | 23.41 ± 0.15 | 9.97 ± 0.44 | 13.43 ± 0.33 | 6.10 ± 0.63 | 32.11 ± 0.16 | 19.53 ± 0.39 | 2.94 |
| | LMA | 11.32 | 9.77 | 24.46 | 9.45 | 19.99 | 8.37 | 11.16 | 5.48 | 29.44 | 16.64 | 2.50 |
| | MA | 10.29 | 9.36 | 23.84 | 8.88 | 21.17 | 8.63 | 12.32 | 5.52 | 30.05 | 17.84 | 2.55 |
| | MiA | 8.16 | 8.22 | 22.77 | 7.95 | 21.97 | 10.35 | 13.37 | 7.20 | 29.92 | 20.57 | 2.77 |
| | S+C | 9.56 | 8.50 | 23.06 | 8.47 | 21.03 | 8.65 | 13.89 | 6.84 | 29.50 | 20.73 | 2.71 |



**Table 4.** Summary of the characteristics of light fraction C isolated from bulk soil and aggregates of ley grassland. A decoupling between SOM dynamics within larger aggregates and fine fractions is highlighted.


| Ley grassland fraction | $^{13}$C stable isotope probing | | $^{13}$C NMR | Interpretation |
|---|---|---|---|---|
| | *Maize-derived material* | *Losses of C3-derived material* | | |
| Bulk | Low (5%) | Low | More advanced status of degradation as for BS | LF-C is shifting through a cropland footprint. |
| LMA | High (30% ≈ PC) | High (PC >PG≈BS) | Rich in fresh plant material | New coarse maize input, untouched by C3 feeding microorganisms |
| MA | Mid (10%) | High (≈ BS) | Low carbohydrates contribution | New maize inputs are low, microorganisms feeds on the scarce C3 remaining |
| MiA | Mid (10%) | Very high | Low carbohydrates and alkyl chains | Scarce C3 source: microbial growth is not sustained |
| S+C | None | Low | Microbial proliferation similar to PG | Remaining C3 material sustains a lower microbial proliferation |



**Figures**

**Figure 1.**


**(a)**

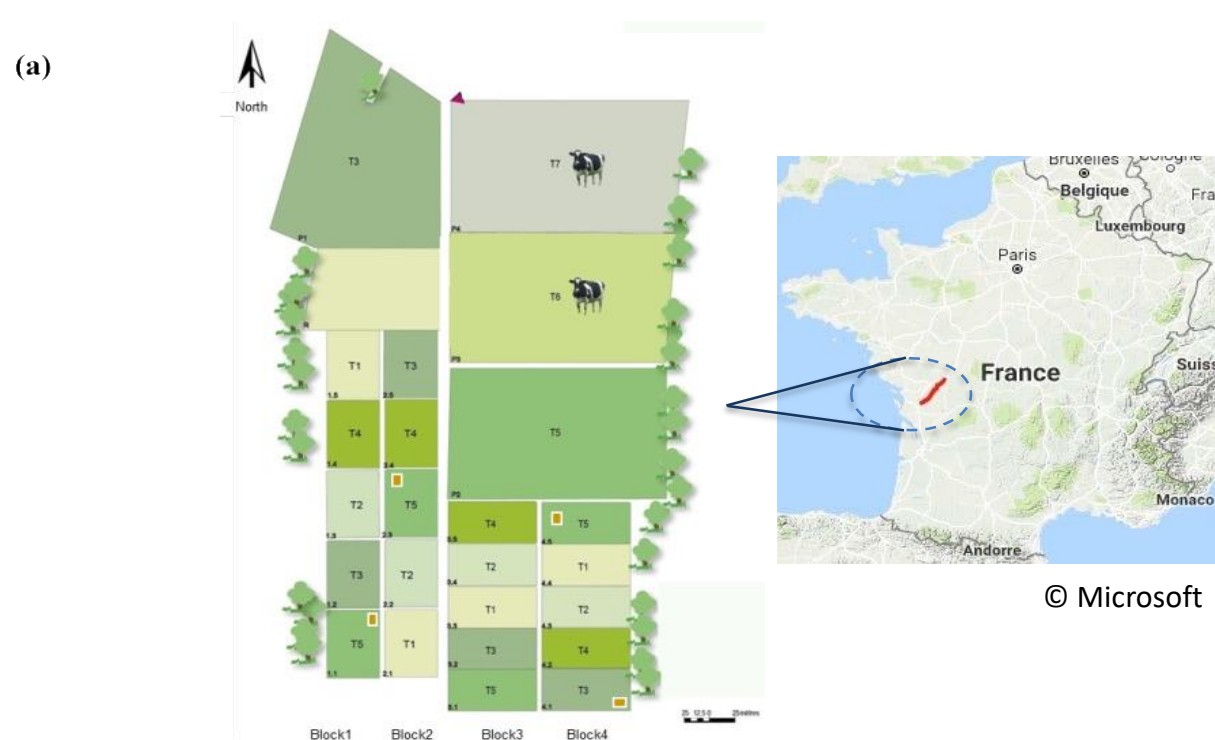

**(b)**

| | | 2005 | 2006 | 2007 | 2008 | 2009 | 2010 | 2011 | 2012 | 2013 |
|---|---|---|---|---|---|---|---|---|---|---|
| T1 | Permanent Cropland | Maize | Maize | Maize | Maize | Maize | Maize | Maize | Maize | Maize |
| T3 | Ley Grassland | Grassland | | | | | | Maize | Maize | Maize |
| T5 | Permanent Grassland | Grassland | | | | | | | | |
| | Bare Fallow | Bare soil | | | | | | | | |





**Figure 2.**

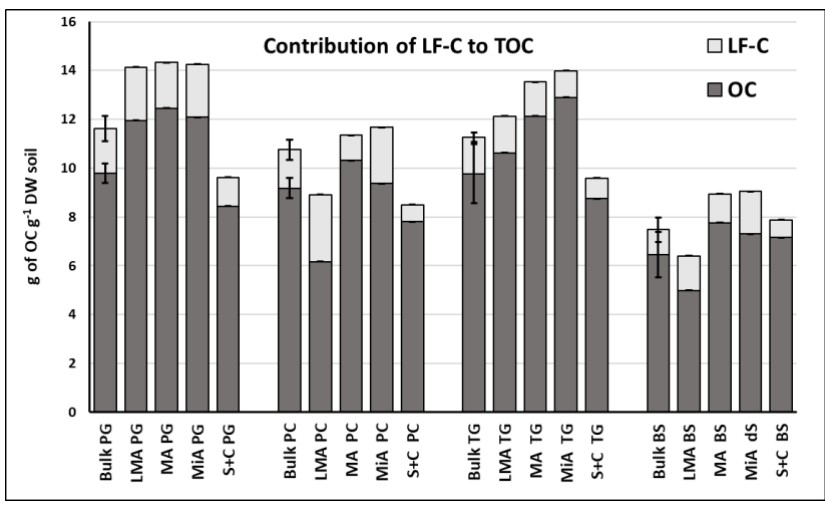

**Figure 3.**


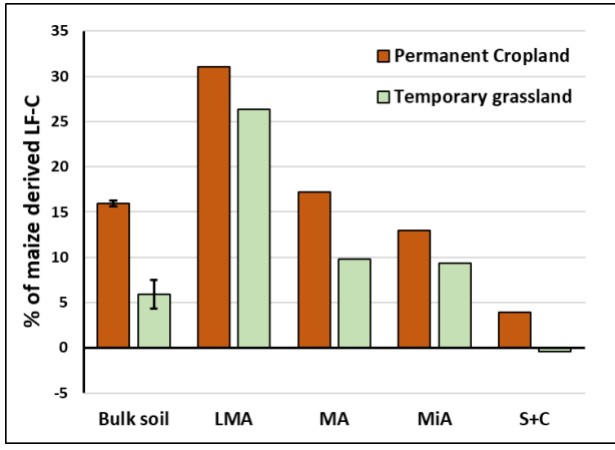

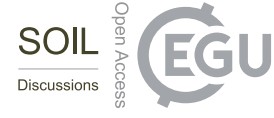

**Figure 4.**

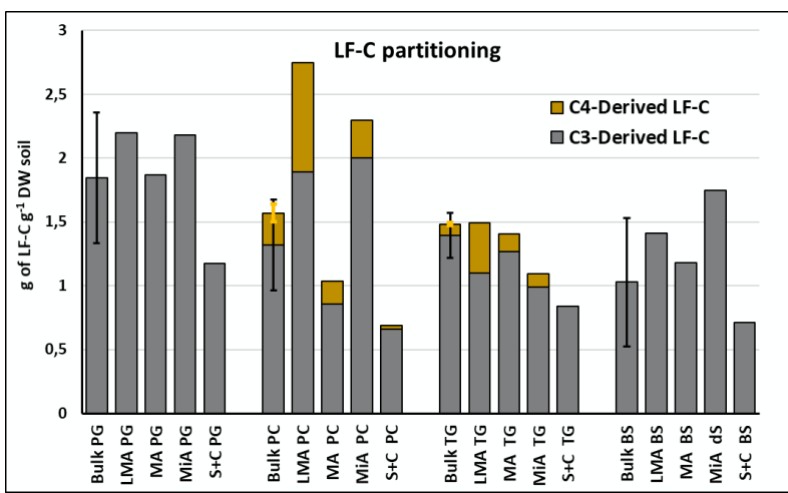


**Figure 5.**

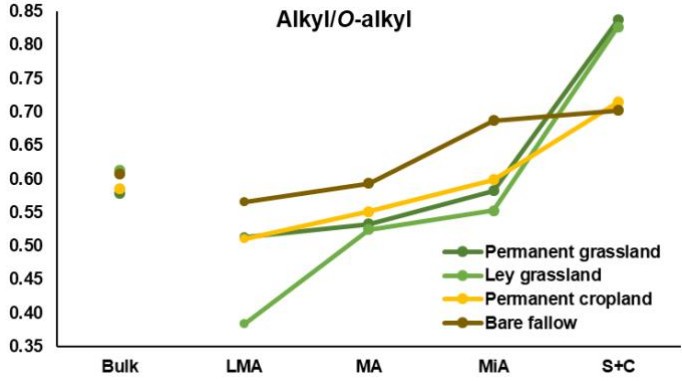

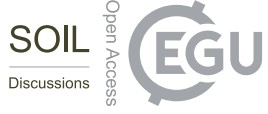

**Figure 6.**


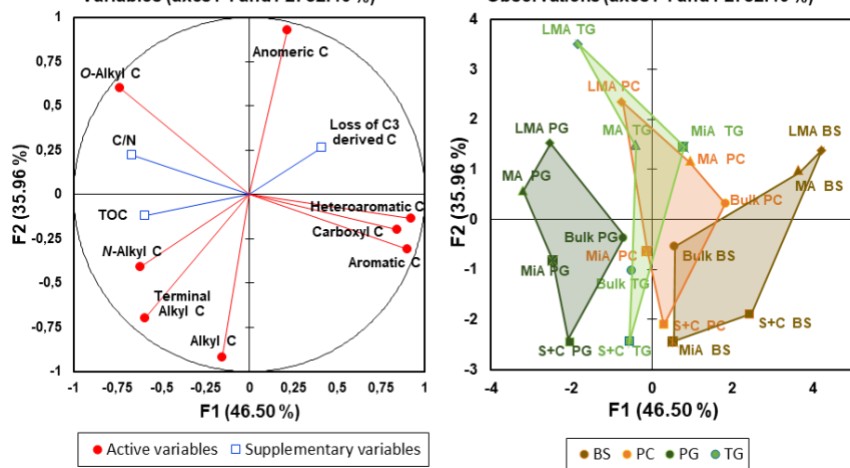



**Figure 7.**

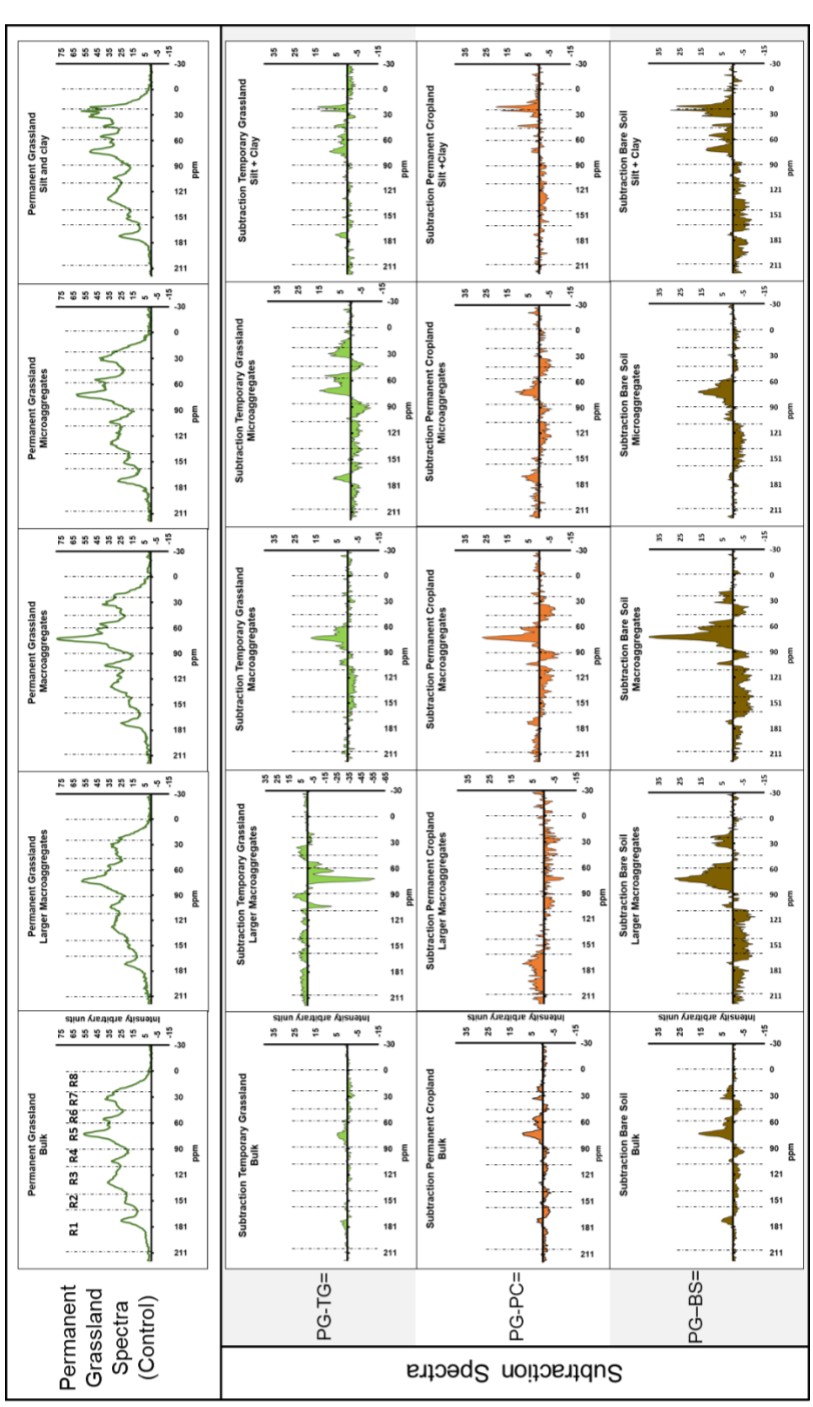