# Peer review of "Land-use perturbations in ley grassland decouple the degradation of ancient soil organic matter from the storage of newly derived carbon inputs."

_SOIL, 2020_

## Referee Comment (RC1) · Anonymous Referee #1 · 7 May 2020

Panettieri et al. have used stable isotope probing and 13C NMR analyses to estimate the evolution of soil C pools in different land use. They focused on the OM light fraction, more sensitive to land use change, and compared their results obtained for four land use: permanent grassland, permanent cropping, ley grassland and bare fallow. The experimental design is very interesting to evaluate land use change effect on OM and especially on C pool isolated by fractionation. This manuscript provides really valuable information on the impact of land use change on OM dynamics and especially the coexistence of to distinct cycle of OM in ley grassland. Only minor modifications should

be made to improve the manuscript. I think that, due to conversion to pdf format, all "13C" have to be checked because they are not in exponent. Similarly, the unit should be in exponent too. The authors used indifferently the terms "temporary grassland" and "ley grassland" (TG or LG) and "bare fallow" and "bare soil" (For example in figures or L346), they should choose one and use only one term. In section 2.1, they use ley grassland (LG) and I think it is the most frequently used in the manuscript. L1- : I think that "on" (focus on) is missing L20 "with grassland returning to soil larger amount of C as belowground inputs than cropping systems": This sentence is not clear. Does it mean that with grassland larger amount of C return to soil as belowground inputs than in cropping systems?

L21 fresh inputs are preferentially incorporated at the level of microaggregates, which are enriched in C in comparison with those of cropped soils: It was not clearly evidenced. For example Figure 4 shows more incorporation of fresh residue in LMA and in figure 2, I am not sure that the difference between aggregate size is significant.

L28 In consequence, vegetal inputs from a new land-use are creating new detritusphere microenvironments rather than sustaining the previous dynamics, resulting in a legacy effect of the previous crop: It is difficult to understand without reading the manuscript. It should be more detailed.

L207 Samples from permanent cropland showed the higher contribution of LF to total stocks of C among the four treatments: It is not so obvious on fig 2. Are the differences significant?

L229 to 233 "under ley grassland and permanent cropland, the MWD was higher for those two treatments if compared with permanent grassland and bare fallow soils": according to table 2, the only significant difference in MWD is between PC and BF. This section should be modified.

L331 exploration of PCA indicated that the type of land-use lead to the highest distances for homologous LMA and MiA fractions: In most of the soils, LMA and S+C

have the highest distances: The authors should explain why they choose LMA and MiA. L327: I agree with the authors, as chemical compounds are more important in bare fallow soil, they could correspond to higher status of degradation of LF. However L329, how do the authors could say that the difference of chemical composition between aggregate size corresponds to degradation status of LF? The difference could reflect different proportion between the OM source : microbial, or maize, or vegetation from grassland.

L338 The fact that mineralization of LF-C from previous land-use was correlated to the N cycle: By previous land use, do you mean grassland? The previous sentence refers to bare soil. I think this sentence should be rephrased to avoid any misunderstanding. Considering my previous comment on OM source in aggregate size fractions, the link between mineralization status and N cycle is not straightforward here. The degradation status in the different fractions should be underpinned.

L349 clearly indicating that LF-C of the treatments under maize presented a more degraded status: I agree but again (CF section 3.4), it is based on the assumption that OM from bare fallow is more degraded. In consequence the authors should clearly present this assumption before, as they did L353.

L381 to 390: I agree with the authors but I think that, in the comparison between PG and PC, rhizodeposition could play an important role. Indeed, as mentioned by the authors in the introduction, L223 section and conclusion, the root traits are very different. But maize provides belowground OM too. The authors should consider this OM source and its effect.

---

## Referee Comment (RC2) · Anonymous Referee #2 · 15 May 2020

1) Introduction – the organization and flow of the introduction needs to be improved. There are short paragraphs that aren't integrated, the objectives are stated before the literature is reviewed in detail. The Introduction section needs major revisions and should have improved logic flow and organization. For example: Line 47: The link of the hypotheses to the literature should be better emphasized. The current structure of the introduction doesn't make it clear how these hypotheses were derived based on research gaps in the literature.

Line 55: This is a short paragraph which should be better integrated with the rest of

the introduction.

Lines 66-67: NMR does not provide such information – clarify.

Line 73: Another hypothesis is stated later in the intro.

2) Methods – all methods seem appropriate. However, it is not justified why only LF was used. This only represents a small portion of the total soil C and analyzing this alone can be misleading. Why were the other fractions not included in some of the analyses in this study? This is a potentially significant limitation becuase mineral associated organic matter (MAOM) provides insight into mechanisms of stabilization and carbon storage.

Lines 92-93 – this information would be more useful if placed in the stable isotope section.

3) Results and Discussion – the organization of this section is very poor. Many short statements with no explanation. Very little data synthesis. The authors need to improve this section for organization and clarity. They must also correct the overinterpretation of the NMR data and be weary about the detection limits of 13C NMR.

This section is also very hard to follow because of the many abbreviations and acronyms used. The authors should revise this entire section carefully and should separate the results and discussion so that the discussion can focus more on what the individual data sets mean when considered holistically. The current format is too fragmented and difficult to follow.

Lines 294-299 and 355 – this isn't correct, a terminal methyl group is not an indication of microbial compounds. Many plant-derived compounds have terminal methyl groups. The authors are misinterpreting the NMR Data here. The NMR data are not resolved enough to provide discreet chemical structures.

Line 345 – it is well documented that the LF is rich in O-alkyl so it is unclear what the point is here.

[Figure]

Line 367 – this is unclear – would changes in vegetation inputs reflect changes in SOM because there is less cutin being added to the soil?

4) Conclusions - because of the poor organization of the R & D section, it is hard to appreciate the conclusions and how the authors made these conclusions based on the data interpretation.

Line 409 – all of the methods have been previously published so there is no novelty in the approach but in the insight.

Tables/Figures Table 3 – there are too many significant figures for the integrated NMR Data. What is the level of reproducibility and detectability? Typically no decimal places are used with such data due to the lack of sensitivity of 13C NMR.

Figure 6 – this figure is very busy and it is unclear what this is showing.

---

## Author Comment (AC1) · 16 May 2020

Dear reviewer,

We would like to thank you for the revision of our manuscript entitled "Land-use perturbations in ley grassland decouple the degradation of ancient soil organic matter from the storage of newly derived carbon inputs."

We have carefully read reviewer's comments and suggestions and we have performed the necessary corrections to the manuscript. In these revision notes the reviewer's

queries are reported in bold letters followed by our answers and comments. We hope that our responses and the changes we made in our manuscript make it suitable for its publication in SOIL.

Sincerely, Dr. Abad Chabbi in behalf of all the co-authors.

Revision notes:

Reviewer comment: Panettieri et al. have used stable isotope probing and 13C NMR analyses to estimate the evolution of soil C pools in different land use. They focused on the OM light fraction, more sensitive to land use change, and compared their results obtained for four land use: permanent grassland, permanent cropping, ley grassland and bare fallow. The experimental design is very interesting to evaluate land use change effect on OM and especially on C pool isolated by fractionation. This manuscript provides really valuable information on the impact of land use change on OM dynamics and especially the coexistence of to distinct cycle of OM in ley grassland. Only minor modifications should be made to improve the manuscript.

Answer: We would like to thank the reviewer for his/her time and for his/her constructive comments. We provide the answers to his comments and concerns and modified the manuscript accordingly.

Reviewer comment: I think that, due to conversion to pdf format, all "13C" have to be checked because they are not in exponent. Similarly, the unit should be in exponent too.

Answer: In fact, a problem arose during the conversion to PDF, we apologize for this inconvenient. We have carefully checked the exponents in this revised version.

Reviewer comment: The authors used indifferently the terms "temporary grassland" and "ley grassland" (TG or LG) and "bare fallow" and "bare soil" (For example in figures or L346), they should choose one and use only one term. In section 2.1, they use ley grassland (LG) and I think it is the most frequently used in the manuscript.

Answer: We adopted the terms "Bare fallow (BF)" and "Ley grassland (LG)" within the text and for all the figures and tables.

Reviewer comment: L1- : I think that "on" (focus on) is missing

Answer: We corrected this sentence.

Reviewer comments: L20 "with grassland returning to soil larger amount of C as belowground inputs than cropping systems": This sentence is not clear. Does it mean that with grassland larger amount of C return to soil as belowground inputs than in cropping systems? L21 fresh inputs are preferentially incorporated at the level of microaggregates, which are enriched in C in comparison with those of cropped soils: It was not clearly evidenced. For example Figure 4 shows more incorporation of fresh residue in LMA and in figure 2, I am not sure that the difference between aggregate size is significant.

Answer: We have completely reworded the sentence at lines 20-23, explaining that belowground inputs are larger for grassland than maize crop under our experimental conditions (as showed in Panettieri et al. 2017 and Armas-Herrera et al. 2016). Furthermore, we removed the part on microaggregates focusing on the fresh maize inputs found in larger macroaggregates. This is to avoid confusion about the names of mid-sized aggregate fractions in the abstract, before the detailed explanation we provided in the manuscript.

Reviewer comments: L28 In consequence, vegetal inputs from a new land-use are creating new detritusphere microenvironments rather than sustaining the previous dynamics, resulting in a legacy effect of the previous crop: It is difficult to understand without reading the manuscript. It should be more detailed.

Answer: We added a more detailed explanation at lines 29-34, as suggested. The new version of the abstract including reviewer's suggestions is now more readable as a stand-alone text.

[Figure]

Reviewer comments: L207 Samples from permanent cropland showed the higher contribution of LF to total stocks of C among the four treatments: It is not so obvious on fig 2. Are the differences significant?

Answer: We reworded the sentence, as requested (actually lines 212-214). Significant differences between treatments of LF-C relative contribution to TOC were not highlighted for the bulk soil samples. Due to the experimental design, we cannot assess significance of the values for the aggregate fractions, but trends to higher relative contribution of LF-C to TOC were found for samples of permanent cropland compared with the other treatments. This is mainly because TOC of aggregates under permanent cropland was lower, but LF-C amount was comparable to the other treatments.

Reviewer comment: L229 to 233 "under ley grassland and permanent cropland, the MWD was higher for those two treatments if compared with permanent grassland and bare fallow soils": according to table 2, the only significant difference in MWD is between PC and BF. This section should be modified.

Answer We modified this section as suggested (now lines 236-238).

Reviewer comment: L331 exploration of PCA indicated that the type of land-use lead to the highest distances for homologous LMA and MiA fractions: In most of the soils, LMA and S+C have the highest distances: The authors should explain why they choose LMA and MiA.

Answer: We were referring to the largest distances between homologous fractions from the different treatments, not between different fractions of the same treatment. We reworded the sentence accordingly (now lines 338-340).

Reviewer comment: L327: I agree with the authors, as chemical compounds are more important in bare fallow soil, they could correspond to higher status of degradation of LF. However L329, how do the authors could say that the difference of chemical composition between aggregate size corresponds to degradation status of LF? The

difference could reflect different proportion between the OM source : microbial, or maize, or vegetation from grassland.

Answer: We agree, we have reworded the sentence as suggested by the reviewer (now lines 335-338).

Reviewer comment: L338 The fact that mineralization of LF-C from previous land-use was correlated to the N cycle: By previous land use, do you mean grassland? The previous sentence refers to bare soil. I think this sentence should be rephrased to avoid any misunderstanding. Considering my previous comment on OM source in aggregate size fractions, the link between mineralization status and N cycle is not straightforward here. The degradation status in the different fractions should be underpinned.

Answer: We have modified the sentence adding a brief explanation on how the litter degradation affect the relative composition of SOM and the redistribution of C pools (now lines 348-350).

Reviewer comment: L349 clearly indicating that LF-C of the treatments under maize presented a more degraded status: I agree but again (CF section 3.4), it is based on the assumption that OM from bare fallow is more degraded. In consequence the authors should clearly present this assumption before, as they did L353.

Answer: We reorganized this section as suggested (now lines 357-360). First we presented the assumption that bare fallow OM is more degraded, then we placed the sentence assessing that OM under maize present a similar degradation pattern for some of the aggregate fractions.

Reviewer comment: L381 to 390: I agree with the authors but I think that, in the comparison between PG and PC, rhizodeposition could play an important role. Indeed, as mentioned by the authors in the introduction, L223 section and conclusion, the root traits are very different. But maize provides belowground OM too. The authors should consider this OM source and its effect.

Answer: We agree, we added a better explanation citing results from Panettieri et al. 2017 and Armas-Herrera et al. 2016 in which the contribution of aboveground and belowground inputs for grassland and maize were evaluated. Of course, maize provides belowground OM, we were referring to the most abundant type of input (Lines 393-400).

Sincerely, Dr. Abad Chabbi in behalf of all the co-authors.
* * *

---

## Author Comment (AC2) · 25 May 2020

We have carefully read reviewer's comments and suggestions and we have performed the necessary corrections to the manuscript.

We hope that our responses and the changes we made in our manuscript make it suitable for its publication in SOIL. Sincerely, Dr. Abad Chabbi in behalf of all the co-authors.

Revision notes:

[Figure]

Reviewer comment: 1) Introduction – the organization and flow of the introduction needs to be improved. There are short paragraphs that aren't integrated, the objectives are stated before the literature is reviewed in detail. The Introduction section needs major revisions and should have improved logic flow and organization. For example: Line 47: The link of the hypotheses to the literature should be better emphasized. The current structure of the introduction doesn't make it clear how these hypotheses were derived based on research gaps in the literature. Line 55: This is a short paragraph which should be better integrated with the rest of the introduction.

Answer: The introduction has been completely reorganized as suggested by the reviewer. Some of the concepts have been clarified, and we improved the integration of all the paragraph of this section. Our first version included a first part about the needs of long-term experimentation in ley grassland rotations, then a second part focused on the technical needs to use early indicators and advanced analytical techniques. This new version presents a merged review of literature followed by the general objectives of our study. This reorganization has improved the logic flow.

Reviewer comment: Lines 66-67: NMR does not provide such information – clarify. Answer: As suggested by the reviewer, we have reworded this sentence referring to the degradation status of SOM induced by land-use or agricultural practices, a proxy that is commonly evaluated using NMR. Reviewer comment: Line 73: Another hypothesis is stated later in the intro.

Answer: We modified the paragraph presenting our hypothesis, as suggested by the

reviewer. Reviewer comment: 2) Methods – all methods seem appropriate. However, it is not justified why only LF was used. This only represents a small portion of the total soil C and analyzing this alone can be misleading. Why were the other fractions not included in some of the analyses in this study? This is a potentially significant limitation becuase mineral associated organic matter (MAOM) provides insight into mechanisms of stabilization and carbon storage.

Answer: We agree that MAOM plays an important role for C stabilization and storage at long term. We have chosen LF due the characteristics explained in the introduction and discussed in the section 3.1. We added a more detailed explanation at lines 69-73.

In summary, land-use changes and crop rotations provides changes in soil C cycles in a shorter timespan than that necessary to observe changes in MAOM. Estimated turnover time of LF is shorter than that of MAOM, thus LF has been proposed in literature as an early-detection pool of C to evaluate land-use or crop rotation performances and to adapt land-use policies for C storage. A previous study on the bulk soil and aggregate fractions from the same experimental area has been published by Panettieri et al. (2017), this new study is focusing on a more reactive pool of soil C and presenting a more detailed chemical characterization.

Reviewer comment: Lines 92-93 – this information would be more useful if placed in the stable isotope section.

Answer: We moved this sentence to the stable isotope section, as suggested.

Reviewer comment: 3) Results and Discussion – the organization of this section is very poor. Many short statements with no explanation. Very little data synthesis. The authors need to improve this section for organization and clarity. They must also correct the overinterpretation of the NMR data and be weary about the detection limits of 13C NMR. This section is also very hard to follow because of the many abbreviations and acronyms used. The authors should revise this entire section carefully and should separate the results and discussion so that the discussion can focus more on what the individual data sets mean when considered holistically. The current format is too fragmented and difficult to follow.

Answer: We reworded some of the sentences to clarify NMR interpretation, as suggested by the reviewer. We have included a supporting reference (Clemente et al., 2011) to justify some of our assumption. We have also reduced the use of acronyms and abbreviations. We have chosen to not separate results and discussion, but we provided a deep reorganization of this section taking into account the suggestions from the two anonymous reviewers. We consider that this new version is less fragmented and it has considerably improved its readability. We think that, considering the high volume and the detail of presented data, a separate section for results will be long and dense and hard to follow without any data discussion associated. However, we are willing to separate this section if the reviewer and/or the editor require it necessary.

Reviewer comment: Lines 294-299 and 355 – this isn't correct, a terminal methyl group is not an indication of microbial compounds. Many plant-derived compounds have terminal methyl groups. The authors are misinterpreting the NMR Data here. The NMR data are not resolved enough to provide discreet chemical structures.

Answer: We have reworded this sentence to clarify the concept. It should be remarked that methoxyl and N-methyl resonates in different spectral regions, and that only terminal methyls bond to methylenes are resonating in the regions we considered. The ratio between terminal methyl and methylene signals has been used to show that the contribution to LF of long-chains attributed to cutins or suberins was not supported by our data. Clemente et al. (2011) used combined techniques of NMR including solution state 1H NMR and diffusion edited 1H NMR to demonstrate that alkyl contribution to fine fractions of soils under prairie is mostly microbial-derived. We included this reference in the manuscript and we also mentioned the possible contribution of plant-derived short aliphatic chains.

Reviewer comment: Line 345 – it is well documented that the LF is rich in O-alkyl so it is unclear what the point is here. Answer: We reworded the sentence to clarify that O-alkyl contribution was higher in samples from permanent grassland than those from the other treatments.

Reviewer comment: Line 367 – this is unclear – would changes in vegetation inputs reflect changes in SOM because there is less cutin being added to the soil?

Answer: We have reworded the sentence making a clear reference to depolymerization

of polysaccharides.

Reviewer comment: 4) Conclusions - because of the poor organization of the R & D section, it is hard to appreciate the conclusions and how the authors made these conclusions based on the data interpretation.

Answer: The reorganized R&D section provides now more connections with the conclusions. Reviewer comment: Line 409 – all of the methods have been previously published so there is no novelty in the approach but in the insight.

Answer: We reworded this sentence as suggested.

Reviewer comment: Tables/Figures Table 3 – there are too many significant figures for the integrated NMR Data.

Answer: We think that the figures we have used are functionally to the chemometrics approach, which extracts more information from the NMR spectra. Figure 5 is functional to show how the alkyl/O-alkyl ratios (common indicators of the degradation status of SOM) are influenced by aggregate-size and land-uses. Figure 6 summarize in a PCA how the spectral regions are correlated with supplementary variables of SOM cycle, and how the PCA plan is grouping the four treatments and the five aggregate fractions on the two principal component axis. Figure 7 summarize the chemometrics approach. It highlights the differences between homologous samples from different treatments. The same results showed as "subtraction spectra" could be represented as histograms or table using the same integration procedure proposed in table 3. However, this figure allows to visually recognize patterns in the spectral regions (e.g. increases in O-alkyl or alkyl intensities) and to distinguish them from the noise/fluctuations. We are willing to reduce the figure numbers if the reviewer and/or the editor consider it necessary.

Reviewer comment: What is the level of reproducibility and detectability?

Answer: About reproducibility: we have tested field replicates for bulk soil samples and reported the standard deviations obtained for each spectral regions (as reported in

the section 3.3) to assure the reproducibility. Working on composite samples for NMR analyses as we did for aggregates is a common approach in literature to cope with time/cost constraints, supported by the reproducibility of NMR analyses (see Diekow et al. 2005, doi: 10.1111/j.1365-2389.2005.00705.x). About detectability: NMR molecular mixing models have been developed taking advantage of other chemical variables (such as C and N contents of samples) and used to give insight on SOM composition (Nelson and Baldock 2004 https://doi.org/10.1007/s10533-004-0076-3 is the most used molecular mixing model). We adopted a similar approach using subtraction spectra and correlation between spectral regions and field variables (total N, 13C isotopic signature or LF-C losses) to highlight main differences in chemical composition of LF-C from aggregates.

Reviewer comment: Typically no decimal places are used with such data due to the lack of sensitivity of 13C NMR.

Answer: As suggested by the reviewer, we reduced to one the decimal the data of table 3, according to the most common format found in recent literature.

Reviewer comment: Figure 6 – this figure is very busy and it is unclear what this is showing.

Answer: Figure 6 is a PCA analysis reducing to two principal components the integrations of the eight NMR spectral regions. In the 6A we represented the active variables (spectral regions) with supplementary variables related to SOM cycle (supplementary variables are not contributing to PCA calculation). In the 6B we represent the distribution of the observations in the plane described by the two principal axes. Observations from the same treatment were connected by polygonal hulls.

---

## Referee Report (RR1)

The revised manuscript by Panettieri et al. has been improved according to reviewers' comments. Some points have been clarified. In consequence, I think that the manuscript could be accepted.

I have only minor comments:

L53 « chemical characterization of SOM will establish C turnover rates": chemical characterization alone cannot provide information on C turnover rate

L86 to 89: the sentence concerning the hypothesis is very long. "stored within different soil compartments": as the focus on LF was questioned by reviewers, the authors should specify here the LF and its indication of early change.

L132: The method of Le Bissonnais is mentioned but I would appreciate some details.

L357: I agree for LG and PG but there is overlap of PC and BF only considering bulk BF.

L396 presumably lost following microbial degradation, rather than from translocation to mineral-associated fraction: nothing supports this hypothesis. I would remove this sentence.

---

## Author Response (AR2)

**SOIL Discuss., [https://doi.org/10.5194/soil-2020-16-RC1](https://doi.org/10.5194/soil-2020-16-RC1)**
*Land-use perturbations in ley grassland decouple the degradation of ancient soil organic matter from the storage of newly derived carbon inputs.*

**Dear SOIL Editorial Board,**
**Dear reviewer,**

We would like to thank you for the second revision of our manuscript entitled "*Land-use perturbations in ley grassland decouple the degradation of ancient soil organic matter from the storage of newly derived carbon inputs.*"

We have performed the necessary corrections to the manuscript suggested by one of the reviewer.
We hope that our responses and the changes we made in our manuscript make it suitable for its publication in SOIL.

*Sincerely,*
*Dr. Abad Chabbi in behalf of all the co-authors.*

**Revision notes**

**Reviewer #1**

**Reviewer comment**: The revised manuscript by Panettieri et al. has been improved according to reviewers' comments. Some points have been clarified. In consequence, I think that the manuscript could be accepted. I have only minor comments:
**Answer:** We would like to thank the reviewer for his/her time and for his/her constructive comments. We provide the answers to his/her minor comments and we have modified the manuscript accordingly.

**Reviewer comment**: L53 « chemical characterization of SOM will establish C turnover rates": chemical characterization alone cannot provide information on C turnover rate.
**Answer:** We modified this sentence. Chemical composition of SOM unveil the degradation patterns, whereas stable isotope probing is used to assess C turnover rate.

**Reviewer comment**: L86 to 89: the sentence concerning the hypothesis is very long. "stored within different soil compartments": as the focus on LF was questioned by reviewers, the authors should specify here the LF and its indication of early change.
**Answer:** We modified the sentence as suggested.

**Reviewer comment**: L132: The method of Le Bissonnais is mentioned but I would appreciate some details.
**Answer:** As requested, we have added more details about Le Bissonnais aggregate fractionation we have used.

**Reviewer comment**: L357: I agree for LG and PG but there is overlap of PC and BF only considering bulk BF.
**Answer:** We agree; we have modified this sentence as suggested.

**Reviewer comment**: L396 presumably lost following microbial degradation, rather than from translocation to mineral associated fraction: nothing supports this hypothesis. I would remove this sentence.
**Answer:** We removed this sentence as requested.

*Sincerely,*

*Dr. Abad Chabbi in behalf of all the co-authors.*